# Modeling of Transmission Compliance and Hysteresis Considering Degradation in a Harmonic Drive

**Ting Tang** [1,2], **Hang Jia** [1,2], **Junyang Li** [1,2,*], **Jiaxu Wang** [1,2] **and Xingyu Zeng** [1,2]

1   College of Mechanical Engineering, Chongqing University, Chongqing 400044, China; tangting0723@cqu.edu.cn (T.T.); myjia@cqu.edu.cn (H.J.); jxwang@cqu.edu.cn (J.W.); 201807021070@cqu.edu.cn (X.Z.)
2   State Key Laboratory of Mechanical Transmissions, Chongqing University, Chongqing 400044, China
*   Correspondence: junyangli@cqu.edu.cn

**Abstract:** Transmission compliance and hysteresis behaviors are associated with harmonic drives. The experimental observation of a harmonic drive showed that these behaviors are significantly degraded with increasing service time, and their accurate modeling is expected to improve the performance of the control system of harmonic drive-based devices such as robot manipulators. In this paper, a new model is proposed to capture transmission compliance and hysteresis and their degradation. The phenomenon is represented through a combination of the nonlinear stiffness component and the micro-sliding friction in the tooth engagement area. The proposed model considered the multi-tooth meshing and the interference effects of the reducer. The parameters of the model were identified using optimization techniques. Numerical simulations and experimental data using a specialized harmonic drive test device were compared to demonstrate the proposed model.

**Keywords:** transmission compliance; hysteresis degradation; interference; multi-tooth meshing

## 1. Introduction

Harmonic drives (HDs), which were invented in the 1950s [1], are well known for their high gear-reduction ratio, low weight, small volume, lack of backlash, and high efficiency [2]. Therefore, HDs are used widely in intelligent robotics and various high-tech fields [3]. Many studies have been conducted on HD kinematic error [4], nonlinear stiffness [5–10], hysteresis [8,10–16], and dynamic friction [17]. However, no definite understanding of the mechanism of transmission compliance and hysteresis behaviors or their degradation is available in the literature. In previous works, the HD is considered to be a black box or grey box, which causes the accuracy of the HD model to degrade over time.

As stated in a manufacturer's catalog [5], the torque-displacement relationship for an HD consists of two characteristic properties: (i) the increasing stiffness with torsional angle, and (ii) the hysteresis characteristic. To capture the nonlinear stiffness, the manufacturer suggests using a piecewise linear function of the torsion angle [5]. Some other scholars prefer to approximate the stiffness curve with a cubic or quintic polynomial [6–8]. Even in certain applications, linear stiffness can be sufficient to simulate the behavior of the reducer [9]. Recently, Zhang et al. [10] proposed the HD rotational compliance model by modeling the compliance behaviors of the flexspline and the wave generator. Considering an HD mechanical system analogy, the transmission compliance is described by the behavior of a serial spring-mass system.

Hysteresis is widespread in the fields of smart materials, magnetic fields, and mechanical systems. The hysteresis in HDs is considered to be caused by the combined effect of torsional flexibility and friction [8,11]. Seyfferth et al. [12] proposed a hysteresis model, represented by a hyperbolic function, in which the hysteresis loss is estimated as a combination of Coulomb friction and a weighted friction function. Assuming that

the hysteresis mainly arose from structural damping, Taghirad [13] developed a model to describe the hysteresis of an HD. Based on the heredity concept of dynamic systems, a mathematically well-posed nonlinear differential equation was proposed by Dhaouadi et al. [8] to describe the hysteresis phenomenon in HDs. Ruderman et al. [14] used the Preisach model to capture the hysteretic behavior of HDs. Preissner et al. [15] proposed a new type of comprehensive harmonic drive model. They used the Maxwell resistive-capacitor model to model the hysteresis behavior in harmonic drives that are taken to be black boxes. Tjahjowidodo et al. [16] developed a grey-box model for HDs that captured the hysteresis behavior by combining nonlinear stiffness characteristics and distributed Maxwell-slip elements. Zhang et al. [10] recently proposed a simple hysteresis model. A hysteresis model considering sheath stiffness was proposed by Hong et al. [18] to capture the hysteretic behavior of the tendon sheath mechanism.

Interferences occur at the contact teeth even at no load in HDs [19]. Sahoo et al. [20] proposed a method of estimating the load shared by the multiple tooth pairs in contact. They pointed out that the interference almost vanished with the increase of the external load. The number of effective meshing teeth pairs increases sharply with the increasing resistance torque [21]. Many experiments also have been done on the lubrication failure of HDs [22–25]. These experimental results showed that obvious wear was found on the wave generator–flexspline interface and the meshing tooth surface. However, these mechanical characteristics of HDs were not considered in the previous grey- or black-box models. This could cause the applicability of the HD model to deteriorate with the degradation of the reducer.

The objective of this study was to develop a mechanical model of transmission compliance and hysteresis in HDs for high-performance robot joint control. The proposed model of transmission compliance and hysteresis was influenced by the work of Tuttle et al. [11] and Tjahjowidodo et al. [16]. Specifically, the proposed mechanical model allows for transmission compliance of the flexspline cup, flexspline teeth, and wave generator; as well as micro-sliding friction in the tooth engagement area, and multi-tooth meshing and interference effects. Our study offers a mechanical model that can capture the degradation of behavior due to wear. A specialized harmonic drive test device was used in the experimental validation of the HD modeling effort.

## 2. Mechanical Model

HDs, which contains only three main components (Figure 1), are widely used in the joints of robot manipulators. The wave generator (WG) is an elliptically shaped steel core surrounded by a flexible race bearing. The flexspline (FS) is a thin-walled flexible cup having two fewer teeth on its outer rim than on the inner rim of the circular spline (CS). The CS is a rigid steel ring with teeth machined into the inner circumference.

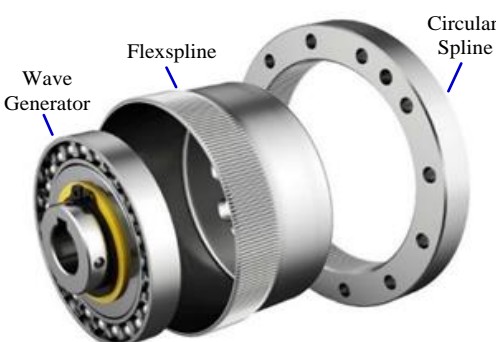

**Figure 1.** The harmonic drive components.

The most common configuration for an HD usually consists of the WG as the input port, the FS as the output port, and the CS fixed to the ground and held immobile. In this configuration, when the WG is inserted into the FS, the FS deforms and causes the FS

teeth and the CS teeth to mesh at the major axis of the WG. With the rotation of the WG, the FS rotates in the opposite direction, corresponding to the load-angle output. Accurate modeling of the HD dynamic behavior is expected to improve the performance of the robot manipulators [16].

### 2.1. Ideal Model

Considering the different working modes for HDs, the ideal kinematic and mechanical relationships of these three ports, as explained in [11], can be expressed as follows:

$$\theta_{WG} = (N+1)\theta_{CS} - N\theta_{FS} \tag{1}$$

$$T_{WG} = \frac{1}{N+1}T_{CS} = -\frac{1}{N}T_{FS} \tag{2}$$

where $\theta_{WG}$, $\theta_{CS}$ and $\theta_{FS}$ denote the angular position of WG, CS, and FS, respectively; $T_{WG}$, $T_{CS}$ and $T_{FS}$ denote the corresponding torques; and $N$ is the gear-reduction ratio, which can be described as:

$$N = \frac{Z_{FS}}{Z_{CS} - Z_{FS}} \tag{3}$$

where $Z_{CS}$ and $Z_{FS}$ denote the number of teeth on the CS and FS, respectively. The most common configuration of an HD is illustrated in Figure 2: a fixed-member CS; a driven-member FS; and a driving-member WG. The basic harmonic-drive equations that capture this three-port behavior are:

$$\theta_{WG} = -N\theta_{FS} \tag{4}$$

$$T_{WG} = -\frac{1}{N}T_{FS} \tag{5}$$

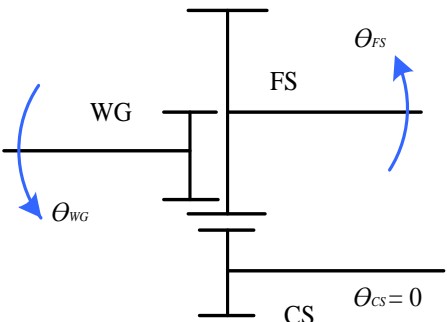

**Figure 2.** A kinematic representation of a harmonic drive showing the three ports.

However, the above equations describe HD behavior under the assumption of an ideal linear input/output relationship, in which friction, torsional compliance, and kinematic error effects are ignored. These factors should be considered to obtain a better result [15].

### 2.2. Basic Model with Friction, Compliance, and Kinematic Error

In order to capture the synthesis effects of friction, torsional compliance, and kinematic error, a model simplified by using a translational motion model analogy, was proposed by Tuttle et al. [11] (Figure 3). In the comprehensive model, the kinematic and mechanical relationships can be expressed as follows:

$$\theta_{WG} = -N(\theta_{FS} + \theta_e + \Delta\theta) \tag{6}$$

$$T_{WG} = -\frac{1}{N}\left(T_{FS} + T_f\right) \tag{7}$$

where $\theta_e$ is the kinematic error, $\Delta\theta$ is the torsional angle of the HD, and $T_f$ is the friction torque. In the model, the compliance behavior is described by modeling the overall behavior of HD.

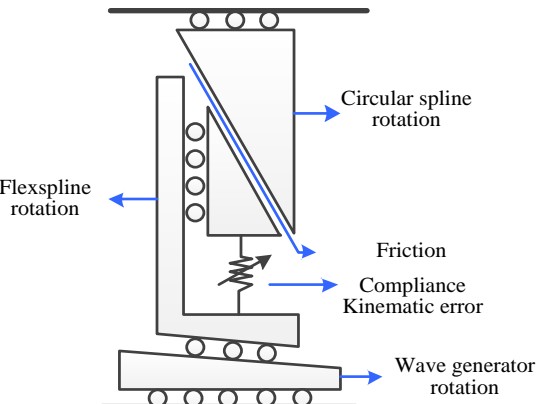

**Figure 3.** A schematic translation representation of the harmonic drive.

According to the description in a manufacturer's catalog [5], the nonlinear spring can be approximated by a piecewise linear function of the torsion angle:

$$\Delta\theta = \begin{cases} \frac{T_{FS}}{K_1} & T_{FS} \leq T_1 \\ \frac{T_1}{K_1} + \frac{T_{FS}-T_1}{K_2} & T_1 \leq T_{FS} \leq T_2 \\ \frac{T_1}{K_1} + \frac{T_2-T_1}{K_2} + \frac{T_{FS}-T_2}{K_3} & T_{FS} \geq T_2 \end{cases} \tag{8}$$

where $K_1$, $K_2$, and $K_3$ are the stiffness; and $T_1$ and $T_2$ are the torque given by the manufacturer. Another method of estimating the nonlinear stiffness uses a cubic polynomial approximation [12]. The third-order polynomial function is given by:

$$T_{FS} = a_3(\Delta\theta)^3 + a_1\Delta\theta \tag{9}$$

where $a_1$ and $a_3$ are constants to be determined.

In order to estimate the friction torque, the Stribeck friction model was used to capture the nonlinear friction behavior [26]. Therefore, the friction torque $T_f$ can be expressed as:

$$T_f = \left[ F_c + (F_s - F_c)e^{-(\dot{\theta}_{WG}/\Omega)^\delta} + F_v\dot{\theta}_{WG} \right] sign\left(\dot{\theta}_{WG}\right) \tag{10}$$

where $F_s$ denotes static friction, $F_c$ denotes the minimum value of the Coulomb friction, $\Omega$ and $F_v$ are lubricant and load parameters, and $\delta$ is an additional empirical parameter.

Any deviation of the teeth from an ideal shape results in the well-known errors in gear ratio known as the kinematic error. As described in [27], the finite Fourier series can be used to describe the kinematic error:

$$\theta_e = \sum_{i=1}^{n} A_i \sin(\omega_i(\theta_{FS}/2\pi) + \Phi_i) \tag{11}$$

where $A_i$ is the amplitude for a particular cyclical frequency, $\omega_i$ is the frequency, $\Phi_i$ is the phase angle for each component, an $n$ value of 3 is adequate to describe the kinematic error profile.

However, as shown in Figure 4, the model described in this section does not incorporate the hysteresis behavior, which results from the interaction between transmission compliance and friction. This is because the model described in this section is a mechanistic superposition model for friction, torsional compliance, and kinematic error, and does not consider the coupling effects of torsional compliance and friction. Hysteresis behavior,

especially in smart materials, magnetic fields, or micro-sliding friction, was dramatically observed [28,29]. A typical HD stiffness curve (Figure 4) is a hysteresis loop that is divided into upper and lower branches. In this study, a new model based on the basic operating principle of HDs is presented to capture their nonlinear torsional compliance and hysteresis behaviors.

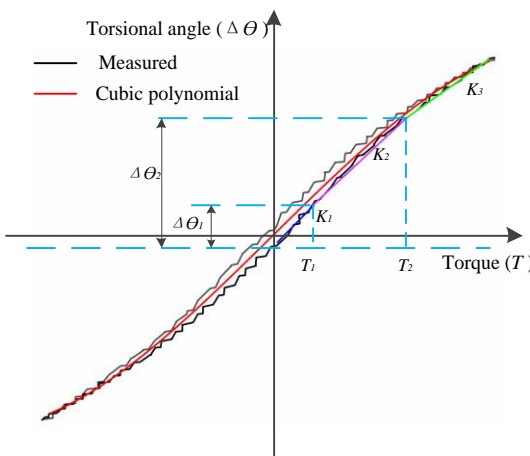

**Figure 4.** A typical stiffness and hysteresis curve of a harmonic drive.

### 2.3. Modeling of Compliance and Hysteresis

Hysteresis behavior is usually caused by micro-sliding friction in mechanical systems [28]. An HD has three moving combinations: between the inner and outer races of the WG bearing, between the outer surface of the WG and the inner surface of the FS, and between the FS and CS teeth. The first two contacting combinations are mainly rolling contacts, and the frictional force is relatively small compared to that of the third sliding combination. Therefore, only the effect of friction in the tooth engagement area on hysteresis was considered. The gear–tooth interface operates in an almost mixed lubrication regime [30]. In this case, the external load was jointly carried by the asperity contact and the fluid film.

$$W = W_l + W_a = (\lambda_1 + \lambda_2)W \qquad (12)$$

where $W_l$ is the load carried by the fluid, $Wa$ is the load carried through direct surface contact, and $\lambda_1$ and $\lambda_2$ are the fluid and asperity load ratio, respectively, which depend on the lubrication condition. The traction force in mixed lubrication can be written as the sum of two portions [31]:

$$F_f = \left(F_f\right)_l + \left(F_f\right)_a = \mu_l W_l + \mu_c W_a \qquad (13)$$

where $(F_f)_l$ and $(F_f)_a$ are the hydrodynamic and asperity traction forces, respectively; $\mu_c$ is the boundary friction coefficient; and $\mu_l$ is the hydrodynamic friction coefficient. Substituting (12) into (13) yields:

$$F_f = (\mu_l \lambda_1 + \mu_c \lambda_2)W = \mu W \qquad (14)$$

In previous works, a single stiffness element was usually included at the HD for simplicity, and the FS and the overall transmission compliance were not quantified into the individual locations of the HD. In practice, the deformation of the WG is also the main contributor to the total transmission compliance [10]. To adequately model HD behavior, in [10], a spring-mass system with two springs and a mass (assume that spring (a) represents the WG compliance, and spring (b) represents the FS compliance) was proposed to describe the stiffness and hysteresis behaviors of the HD transmission. It implied that the compliance of the WG needs to be considered when modeling the HD compliance.

Previous researchers have observed that the number of teeth pairs in an HD that mate simultaneously increased with increasing resistance torque [21]. This is because

the deflections of the FS and CS teeth gradually increase with the increasing load torque. This shows that the FS compliance is composed of the compliance of the FS cup and the FS teeth. The CS teeth are much more rigid than the FS teeth in conventional HDs. Therefore, the CS teeth contribution to transmission compliance was not considered in this paper. The stiffness of an FS tooth is expressed as follows:

$$K_{Ti} = \frac{F_i}{\zeta} \tag{15}$$

where $F_i$ is the gear loads, and $\zeta$ is the total deflection of an FS tooth. The total deflection is composed of deflections due to bending, deflections due to shearing, local deflections owing to Hertzian contact deformation, and deflections due to tooth foundation [20].

In particular, interferences occur at several tooth pairs even with no load, but the interference almost vanishes with an increase in the external load [20]. The no-load preload $T_0$ due to inherent initial interference can be approximated by a piecewise linear function of the load:

$$T_0 = \begin{cases} c(T_{FS} - T_i) & T_{FS} \le T_i \\ 0 & T_{FS} > T_i \end{cases} \tag{16}$$

where $c$ is constant to be determined, and $T_i$ is the piecewise linear threshold.

Based on the above discussions, we observed that the main sources of flexibility in HD transmissions included the FS cup, the FS teeth, and the WG. The hysteresis phenomenon resulted from the combined effect of the flexibility in the HD and micro-sliding friction in the tooth engagement area. In order to model the HD hysteresis, we considered its mechanical system analogy, and an HD model was proposed by using a translational motion model analogy (Figure 5). The behavior of the HD rotational compliance can be captured by the behavior of the spring and massless inclined plane system. The aggregate effect of multiple gear teeth meshing is instead achieved by a single continuous gear-tooth surface. As illustrated in Figure 5, $K_F$, $K_T$, and $K_W$ represent the FS cup, FS teeth, and WG compliance, respectively; $T_f$ is the HD friction torque; and $T_0$ is no-load preload. When $T_{FS}$ increased from zero, springs $K_F$ and $K_T$ were compressed, the number of working springs $K_T$ increased, the no-load preload $T_0$ decreased, and spring $K_W$ stayed at its initial position as long as the driving force was less than friction torque $T_f$.

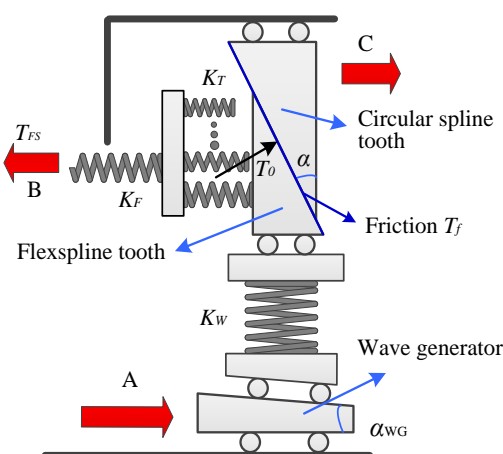

**Figure 5.** A harmonic drive model with gear-tooth geometry.

In order to model the FS teeth compliance (Figure 5), a series of springs with different lengths and stiffnesses were used to embody the FS teeth of the actual transmission. Varying spring lengths were intended to characterize the fact that the number of meshing tooth pairs varies with load. FS tooth stiffness was influenced by two main factors: the gear tooth geometry and parameters, and the point of application of load. Pedersen and Jorgensen [32] showed that the stiffness had an almost parabolic dependence on the contact point position.

The combined stiffness of meshing tooth pairs can be simplified by a piecewise linear function of the load:

$$K_T = \begin{cases} aT_{FS} + b & T_{FS} \leq T_T \\ aT_T + b & T_{FS} > T_T \end{cases} \tag{17}$$

where $a$ and $b$ are constants to be determined, and $T_T$ is the piecewise linear threshold.

From the analysis of the HD model in Figure 5, the torque-displacement relationship of the HD gear can be modeled as

$$T_{FS} = K_F \Delta\theta_F \tag{18}$$

$$\Delta\theta_T \approx \begin{cases} \sum\limits_{i=1}^{li(T_{FS})} \frac{1}{ai+b} & T_{FS} \leq T_T \\ \sum\limits_{i=1}^{li(T_T)} \frac{1}{ai+b} + \frac{T_{FS}-T_T}{aT_T+b} & T_{FS} > T_T \end{cases} \tag{19}$$

$$T_{FS}\sin\alpha = K_W\Delta\theta_W\cos\alpha \pm (T_{FS}\cos\alpha + K_W\Delta\theta_W\sin\alpha + T_0)\mu \tag{20}$$

where $\Delta\theta_W$, $\Delta\theta_T$, and $\Delta\theta_F$ denote the torsional angles of WG, FS teeth, and FS cup, respectively; $\alpha$ is the gear-tooth angle, and $li(T_{FS})$ and $li(T_T)$ are the largest integers less than $T_{FS}$ and $T_T$, respectively. Equation (20) incorporates information about the direction-dependent friction torque and no-load preload.

When considering the compliance of the WG, the FS teeth, and the FS cup, according to the basic operating principle shown in Figure 5, the torsional angle of the HD can be obtained by using the following expression:

$$\Delta\theta = \Delta\theta_F + \Delta\theta_T + \tan\alpha \cdot \Delta\theta_W \tag{21}$$

## 3. Behavioral Degradation

In order to evaluate the behavioral degradation, a specialized harmonic drive test device was designed to observe the behavior change of the reducer. As depicted in Figure 6, it is an electromechanical system that includes a driving servo motor, a loading servo motor, two rotary encoders, two torque sensors, an HD unit, and a digital control module. The concentricity was adjusted using an Easy-Laser shaft alignment instrument. The main performance parameters of the test apparatus are displayed in Table 1. An HD (CSF-25-80) was used for the behavioral degradation test. The reduction ratio was 80:1, and it had a rated output torque of 63 Nm.

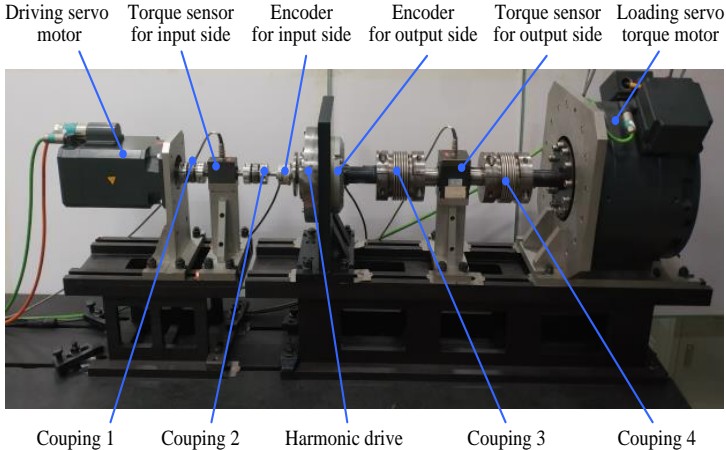

**Figure 6.** View of the harmonic drive test apparatus.

To determine the torsional compliance and hysteresis behaviors of the HD experimentally, first, the input shaft of the drive was locked. Next, the load torque was controlled to vary in a linear manner, and the resulting angular displacement for the load position was

measured by the rotary encoder and recorded. The initial torque-displacement relationship of the reducer is shown in Figure 7. Then the reducer was operated stably for 100 h at a rated torque of 63 Nm and a rated speed of 2000 rpm, and its torque-displacement relationship was tested (Figure 7). The wear conditions of the rubbing surface of the HD unit after the test are shown in Figure 8. The wear was visible on the FS tooth surface and FS inner surface.

**Table 1.** List of the main performance parameters of the test system.

| Name | Rating | Resolution |
| --- | --- | --- |
| Driving servo motor | 6.5 Nm | - |
| Torque sensor for the input side | 20 Nm | 0.02 Nm |
| Encoder for input side | - | 0.00439° |
| Loading servo torque motor | 500 Nm | 0.5 Nm |
| Torque sensor for the output side | 1000 Nm | 0.1 Nm |
| Encoder for the output side | - | 0.00056° |

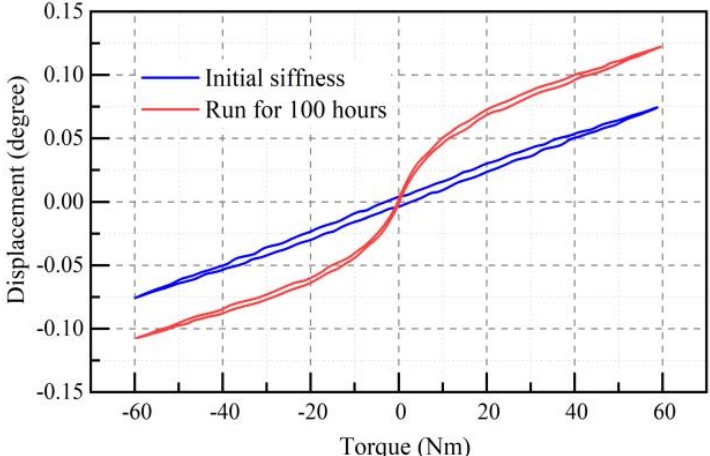

**Figure 7.** The transmission compliance and hysteresis behaviors change of the HD.

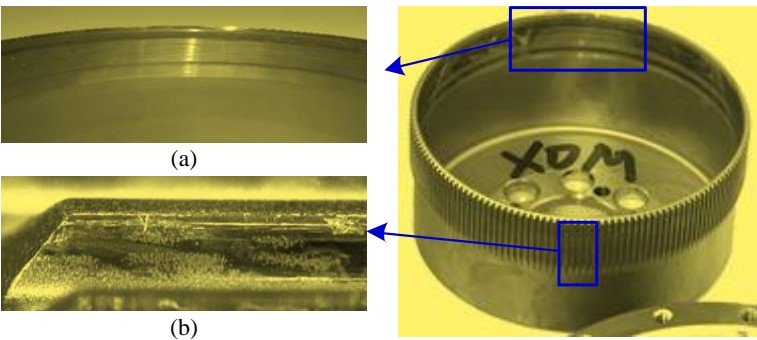

**Figure 8.** The condition of the rubbing surface after the test: (**a**) FS inner surface, and (**b**) FS tooth.

Figure 7 shows that behavioral degradation of the reducer occurred. Therefore, in order to improve the performance of the system, it was necessary to capture the degradation behavior of the reducer. Using the analysis of the HD model in Figure 5, the transmission compliance and hysteresis in HD were mainly attributed to the flexibility of the FS cup, the FS teeth, and the WG, as well as the friction in the tooth engagement area. The degradation of the behavior of the reducer was caused by the wear of the FS tooth surface and the FS inner surface, which led to the degradation of the stiffness of the FS tooth and the friction in the tooth engagement area.

For the FS tooth, wear made the tooth and tooth foundation thinner, and reduced the depth of engagement and number of meshing teeth, which led to degradation of the combined stiffness of meshing tooth pairs.

$$K'_T = \begin{cases} a'T_{FS} + b' & T_{FS} \le T'_T \\ a'T'_T + b' & T_{FS} > T'_T \end{cases} \tag{22}$$

At the same time, with the increase of wear, the amount of interference of the tooth surface gradually decreased, and when the wear reached a certain degree, the interference disappeared. Therefore, the no-load preload $T_0$ due to inherent interference also decreased with the increase of wear until it reached 0.

$$T_0' = \begin{cases} c'(T_{FS} - T') & T_{FS} \le T_i' \\ 0 & T_{FS} > T_i' \end{cases} \tag{23}$$

Equations (22) and (23) capture the degradation of this behavior in the proposed model. Figure 7 shows that the initial transmission compliance and hysteresis in HD had the symmetric property, but the reducer did not have such properties after 100 h of operation. This is most likely because the test was conducted in one direction. Figure 7 also shows that the reducer hysteresis at zero load torque decreased and tended toward zero after 100 h of testing. This is because the tooth surface interference disappears due to wear, and the no-load preload is zero. Therefore, the micro-sliding friction that caused hysteresis is almost zero at no load. Furthermore, the transmission compliance shows obvious nonlinearity, especially in the range of small-load torque.

## 4. Parameter Identification

The steady-state hysteresis loop can be defined by three curves [8]: the stiffness curve, an ascending curve denoted for increasing load ($dT_{FS}/dt > 0$), and a descending curve denoted for decreasing load ($dT_{FS}/dt < 0$). According to the model proposed in this paper, the stiffness curve can be obtained by using the following expression:

$$\Delta\theta = \left(\frac{1}{K_F} + \frac{\tan^2(\alpha)}{K_W}\right)T_{FS} + \Delta\theta_T(T_{FS}) \tag{24}$$

From the analysis of the HD model in Figure 5, the torque-displacement relationship of the HD gear can be represented by two functions. For an increasing load ($dT_{FS}/dt > 0$), the function will take the form:

$$\Delta\theta_u = \frac{T_{FS}}{K_F} + \Delta\theta_T(T_{FS}) + \tan\alpha \cdot \Delta\theta_{Wu} \tag{25}$$

$$\Delta\theta_{Wu} = \begin{cases} -\frac{\mu(T_0 - T_{FS}\cos\alpha) - T_{FS}\sin\alpha}{K_W(\cos\alpha - \mu\sin\alpha)} & T_{FS} \le \frac{T_o\mu}{\sin\alpha + \mu\cos\alpha} \\ 0 & \frac{T_o\mu}{\sin\alpha + \mu\cos\alpha} < T_{FS} \le \frac{T_o\mu}{\sin\alpha - \mu\cos\alpha} \\ \frac{T_{FS}(\sin\alpha - \mu\cos\alpha) - T_0\mu}{K_W(\cos\alpha + \mu\sin\alpha)} & T_{FS} > \frac{T_o\mu}{\sin\alpha - \mu\cos\alpha} \end{cases} \tag{26}$$

For a decreasing load ($dT_{FS}/dt < 0$), the function will take the form:

$$\Delta\theta_d = \frac{T_{FS}}{K_F} + \Delta\theta_T(T_{FS}) + \tan\alpha \cdot \Delta\theta_{Wd} \tag{27}$$

$$\Delta\theta_{Wd} = \begin{cases} \frac{T_{\max}(\sin\alpha - \mu\cos\alpha) - T_0(T_{\max})\mu}{K_W(\cos\alpha + \mu\sin\alpha)} & T_{FS} \ge \frac{K_W\Delta\theta_{W\max}(\cos\alpha - \mu\sin\alpha) - T_0\mu}{\sin\alpha + \mu\cos\alpha} \\ \frac{T_{FS}(\sin\alpha + \mu\cos\alpha) + T_0\mu}{K_W(\cos\alpha - \mu\sin\alpha)} & T_{FS} < \frac{K_W\Delta\theta_{W\max}(\cos\alpha - \mu\sin\alpha) - T_0\mu}{\sin\alpha + \mu\cos\alpha} \end{cases} \tag{28}$$

According to the model proposed in this paper, the nonlinearity of transmission compliance is caused by the FS teeth. When $T_{FS} > T_T$, the rigidity of the FS tooth is constant, the transmission flexibility of the reducer is linear. Therefore, the threshold value $T_T$ can

be obtained by the method shown in Figure 9. A trial and error method can be used to estimate the parameters of the proposed HD model. The $\mu$ and $cT_i$ take initial values as $\mu_0$ and $(cT_i)_0$. As shown in Figure 9, five points ($A_i$ ($A_{i1}$, $A_{i2}$) $i = 1, \ldots, 5$) are selected for parameter evaluation. Considering the test error, the average value can be obtained through several tests. Model parameters can be estimated as follows:

$$\hat{K}_W = \frac{\mu_0(cT_i)_0 \tan \alpha}{A_{12}(\cos \alpha - \mu_0 \sin \alpha)} \tag{29}$$

$$\hat{c} = \frac{A_{32}\hat{K}_W(\cos \alpha - \mu_0 \sin \alpha)}{2\mu_0 A_{31} \tan \alpha} + \frac{(cT_i)_0}{A_{31}} \tag{30}$$

$$\hat{T}_i = \frac{(cT_i)_0}{\hat{c}} \tag{31}$$

$$\frac{A_{41}}{\hat{K}_F} + \Delta\hat{\theta}_T(A_{41}) + \tan \alpha \Delta\hat{\theta}_{Wu}(A_{41}) = A_{42} \tag{32}$$

$$\frac{A_{51}}{\hat{K}_F} + \Delta\hat{\theta}_T(A_{51}) + \tan \alpha \Delta\hat{\theta}_{Wu}(A_{51}) = A_{52} \tag{33}$$

$$\frac{A_{21}}{\hat{K}_F} + \Delta\hat{\theta}_T(A_{21}) + \tan \alpha \Delta\hat{\theta}_{Wu}(A_{21}) = 0 \tag{34}$$

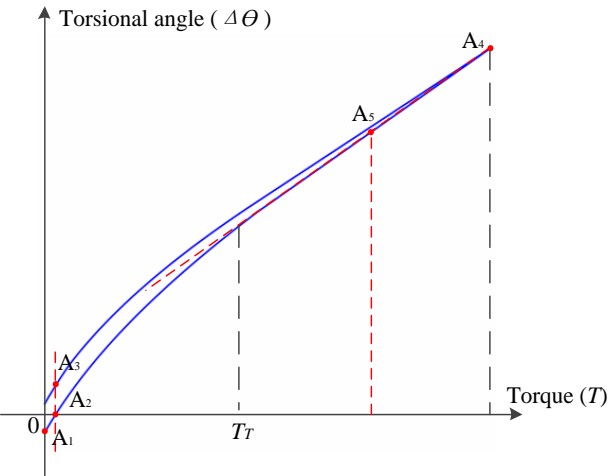

**Figure 9.** The stiffness and hysteresis curve of the spring and massless inclined plane system.

Based on estimated stiffness parameters ($K_F$, $K_W$, $a$, $b$, $T_T$) and hysteresis parameters ($c$, $T_i$, $\mu$), the evaluation errors $\varepsilon_{uj}$ $\varepsilon_{dj}$ can be expressed as follows:

$$\Delta\theta_{uj} = \Delta\hat{\theta}_{uj} + \varepsilon_{uj} \tag{35}$$

$$\Delta\theta_{dj} = \Delta\hat{\theta}_{dj} + \varepsilon_{dj} \tag{36}$$

where $\Delta\theta_{uj}$ and $\Delta\theta_{uj}$ are the measured hysteresis torsional angles. The optimum parameters $\mu_0$ and $(cT_i)_0$ in the least square sense can be obtained to minimize the criterion function in Matlab software:

$$J = \sum_{j=1}^{n} \left( \varepsilon_{uj}^2 + \varepsilon_{dj}^2 \right) \tag{37}$$

Specifically, when $T_T = 0$, the torsional angle of FS cup and FS teeth can be rewritten as follows:

$$\frac{T_{FS}}{K_F} + \Delta\theta_T(T_{FS}) = \frac{T_{FS}}{K} \tag{38}$$

where $K$ is the combined stiffness of the FS cup and FS teeth, which can be evaluated using the above parameter-evaluation method.

## 5. Results and Discussion

Considering the non-symmetric property of the HD's stiffness after testing, the transmission compliance and hysteresis model parameters were identified using the torque-displacement curves for $T_{FS} > 0$ (Figure 7). The identified parameters are listed in Table 2.

**Table 2.** Experimentally identified parameter values used in the HD simulation.

| Parameter | Initial Curve | Post-Test | Parameter | Initial Curve | Post-Test |
|---|---|---|---|---|---|
| $T_T$ | 0 Nm | 30 Nm | $K_W$ | 101,837.5 Nm/rad | 98,548.7 Nm/rad |
| $A$ | - | 1833.5 | $M$ | 0.066 | 0.076 |
| $B$ | - | 5729.6 | $C$ | −1.177 | −1.152 |
| $K_F$ (K) | 50,923.2 Nm/rad | 78,197.8 Nm/rad | $T_i$ | 115.5 Nm | 70.2 Nm |

The torsional angle was computed using the identified parameters and the transmission compliance and hysteresis model as given in Equations (25)–(28). The estimated torsional angle was next compared to the actual measured angle. Figure 10 shows the estimated and measured torque-displacement curves for different run times. The calculated curves closely coincided with those found experimentally. Therefore, the spring and massless inclined plane system can reasonably describe the transmission compliance and hysteresis behaviors of the HD.

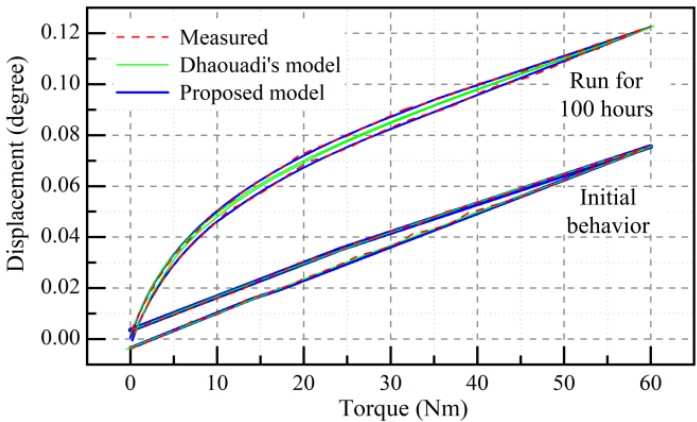

**Figure 10.** The measured and simulated hysteresis curves.

A dynamic model of hysteresis in HDs proposed by Dhaouadi et al. (Appendix A) also was used to describe the transmission compliance and hysteresis phenomenon. As shown in Figure 10, when comparing the proposed model to the Dhaouadi's model, for the initial behavior, the estimation results formed two similar curves that are consistent with the experimental observation. However, for the HD after 100 h of operation, the model proposed in this paper can better characterize its hysteresis behavior. This is because the proposed model captures the behavior degradation mechanism of the reducer.

The torque-displacement curve of the Zhang's hysteresis model given in Appendix B, the Dhaouadi's hysteresis model given in Appendix A, and that of the proposed model are shown in Figure 11. In Zhang's model, the behavior of the HD's rotational compliance was described by the behavior of a serial spring-mass system. They assumed that the linear spring (a) represents the WG compliance, the linear spring (b) represents the FS compliance, and hysteresis is caused by constant friction $F_f$. It can be shown that Zhang's model cannot capture the nonlinear flexibility of a harmonic drive, and the hysteresis curve is independent of the load torque. In Dhaouadi's model, they used the hereditary concept to

represent the hysteresis behavior of the HD, and an analytical odd polynomial function was selected for the stiffness curve. It is important to observe that the hysteresis represented by this model decreased with an increase in load. Therefore, as shown in Figure 10, this model cannot accurately capture the hysteretic behavior of the harmonic reducer after degradation. Compared with the above models, the proposed model considers the influence of the multi-tooth engagement and interference characteristics of the HD on the transmission flexibility and hysteresis. The proposed model can capture the degradation of nonlinear stiffness and hysteresis behaviors of HD.

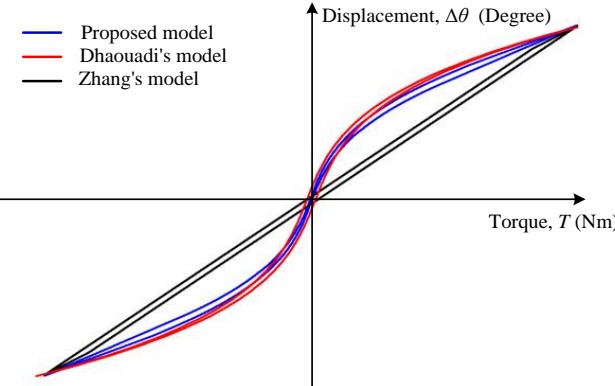

**Figure 11.** A comparison of the proposed model and the existing model.

Figure 12 shows an analysis of the effects of interference and multi-tooth meshing on the behavior of the HD. The hysteresis was clearly affected by the preload caused by the interference. At zero torque output, hysteresis decreased as preload decreased. This explains why the hysteresis decreased as the running time increased (Figure 7). Figure 12 also shows that the nonlinear stiffness of the reducer is attributed to the number of meshing teeth, which depends on the load. The number of meshing teeth reached the maximum ($T_T = 0$) due to initial interference. Therefore, the initial stiffness was linear (Figure 7). With an increase in wear, the initial no-load tooth interference vanishes gradually, and $T_T$ increased from zero. The transmission compliance of HD shows nonlinear characteristics gradually. This is consistent with the test results shown in Figure 7.

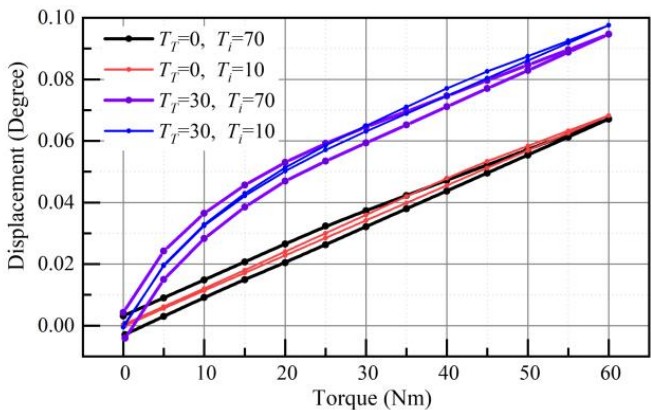

**Figure 12.** The influence of interference and multi-tooth engagement on the transmission compliance and hysteresis.

Figure 13 shows the analysis of the effects of the friction coefficient and stiffness of the FS and the WG on the behavior of the HD. With the increase of the friction coefficient, the hysteresis characteristic strengthened, and the stiffness of the reducer increased slightly. The decrease in the stiffness of the WG not only increased the hysteresis characteristics of the reducer, but also reduced the stiffness of the reducer. For the FS stiffness, when it

was reduced, it only reduced the overall rigidity of the reducer, and hardly affected the hysteresis characteristics of the reducer.

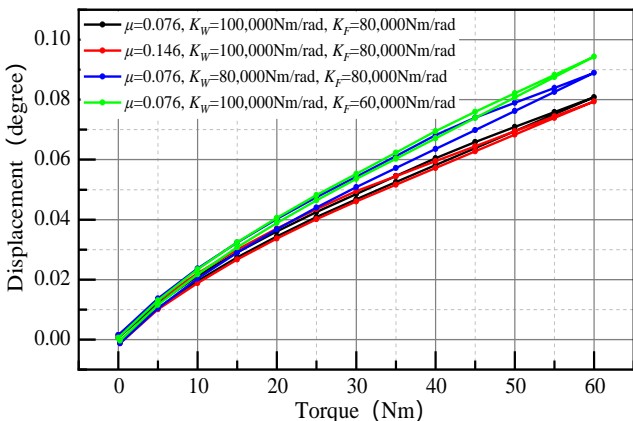

**Figure 13.** The influence of the friction coefficient and FS and WG stiffness on the transmission compliance and hysteresis.

The derived stiffness model given in Equation (24) was compared with the linear piecewise model given in Equation (8) and the cubic polynomial model given in Equation (9) through simulations. Figure 14 shows the simulation results for the piecewise linear model, cubic polynomial model, and proposed model. The results show that the proposed stiffness model matched the others. Figure 15 shows the local elastic coefficient evaluated using different models. The local elastic coefficient can be calculated as:

$$K_{FL} = \frac{\mathrm{d}T_{FS}}{\mathrm{d}\Delta\theta} \tag{39}$$

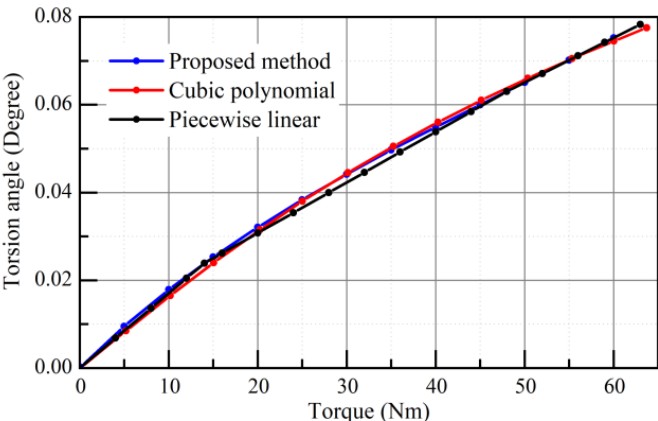

**Figure 14.** The stiffness curves for the piecewise linear model, cubic polynomial model, and proposed model.

In Figure 15, the proposed stiffness model's local elastic coefficient curve consists of two parts. When $T_{FS} < T_T$, the local elastic coefficient, similar to those achieved using the cubic polynomial model, increased as $T_{FS}$ increased. When $T_{FS} > T_T$, the local elastic coefficient remained unchanged, which is consistent with the results of the piecewise linear model. It can be seen in Figure 7 that the above properties of the proposed model captured the stiffness degradation characteristics of the reducer. The maximum relative error of the proposed model and the cubic polynomial model was 26.3%, and that of the proposed model and the piecewise linear model was 24.8%.

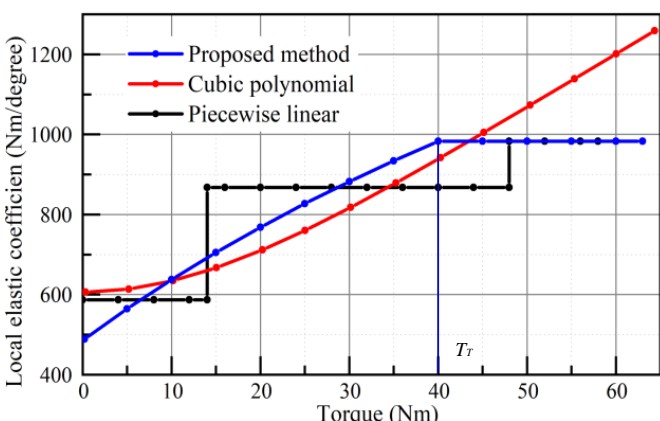

**Figure 15.** The simulation results for the local elastic coefficient.

## 6. Conclusions

In this study, we presented a new model based on the mechanical system analogy to explain the transmission compliance and hysteresis behaviors in HDs. By considering the effect of interference and multi-tooth meshing, the proposed model captured the degradation of reducer behavior caused by wear. The hysteresis loss was captured by taking the micro-sliding friction in the tooth engagement area into account. The main sources of flexibility in HD transmissions include the FS cup, the FS teeth, and the WG. The number of meshing teeth was affected by the load torque, which is the main cause of nonlinear stiffness. The interference, WG stiffness, and friction coefficient were the main influencing factors in hysteresis behavior. The method of estimating the parameters of the proposed HD model has been described in detail. Experimental results confirmed the effectiveness of the proposed model.

Furthermore, the model can capture the degradation of transmission flexibility and hysteresis better than the existing models. Therefore, the model will be useful in understanding the hysteresis mechanism of HDs. There are two characteristics of the behavioral degradation of HD: the degradation of stiffness and enhancement of nonlinear characteristics, and the reduction of hysteresis loss near zero load.

**Author Contributions:** Funding acquisition—J.L., J.W.; project administration—J.L., J.W.; conceptualization —T.T., H.J.; validation—X.Z.; formal analysis—T.T., H.J.; investigation—T.T., J.L.; data curation—H.J., X.Z.; writing—original draft preparation—T.T.; writing—review and editing—T.T. All authors have read and agreed to the published version of the manuscript.

**Funding:** This work was supported by the National Key Research and Development Program of China (2018YFB1304800) and Key Research and Development Program of Guangdong Province (2020B090926002).

**Institutional Review Board Statement:** Not applicable.

**Informed Consent Statement:** Not applicable.

**Data Availability Statement:** The data presented in this study are available on request from the corresponding author.

**Conflicts of Interest:** The authors declare no conflict of interest.

**Abbreviations**

| | |
|---|---|
| $N$ | reduction Ratio |
| $Z_{CS}$, $Z_{FS}$ | teeth number of CS and FS |
| $\theta_e$ | kinematic error |
| $\Delta\theta$ | harmonic drive torsional angle |
| $\theta_{WG}$, $\theta_{CS}$, $\theta_{FS}$ | angular position of WG, CS, and FS |
| $\Delta\theta_W$, $\Delta\theta_T$, $\Delta\theta_F$ | torsional angle of WG, FS teeth, and FS cup |
| $\Delta\theta_u$, $\Delta\theta_d$, | harmonic drive torsional angle for increasing and decreasing load |
| $\Delta\theta_{uj}$, $\Delta\theta_{dj}$ | estimated hysteresis torsional angle |
| $\Delta\theta_{Wu}$, $\Delta\theta_{Wd}$, | wave generator torsional angle for increasing and decreasing load |
| $T_{WG}$, $T_{CS}$, $T_{FS}$ | torque of WG, CS, and FS |
| $T_1$, $T_2$ | torque given by the manufacturer |
| $T_f$ | friction torque |
| $T_0$ $(T_0{}')$ | no-load preload |
| $T_i$ $(T_i{}')$ | piecewise linear threshold |
| $K_T$ $(K'_T)$ | combined stiffness of meshing tooth pairs |
| $T_T$ $(T_T{}')$ | piecewise linear threshold |
| $K_{FL}$ | local Elastic Coefficient |
| $K_F$, $K_T$, $K_W$ | stiffness of FS Cup, FS Teeth, and WG |
| $K_1$, $K_2$, $K_3$ | stiffness Given by the Manufacturer |
| $F_s$ | static friction |
| $F_c$ | minimum value of the Coulomb friction |
| $F_v$ | load parameter |
| $\Omega$ | lubricant parameter |
| $A_i$ | amplitude for a particular cyclical frequency |
| $\Phi_i$ | phase angle |
| $\omega_i$ | frequency |
| $a$, $b$($a'$, $b'$) | coefficient of combined stiffness |
| $c$ $(c')$ | coefficient of no-load preload |
| $\alpha$ | gear-tooth angle |
| $\delta$ | additional empirical parameter |
| $\mu_c$, $\mu_l$ | boundary and hydrodynamic friction coefficient |
| $\mu$ | friction coefficient |
| $W_l$, $W_a$ | load carried by the fluid and asperity |
| $\lambda_1$, $\lambda_2$ | fluid and asperity load ratio |

**Appendix A. Dhaouadi's Hysteresis Model**

The hysteresis loop consists of two curves:

$$h_u(t) = f(\Delta\theta(t)) - \frac{A}{\beta}\left(-1 + 2\frac{e^{-\beta(Q+\Delta\theta(t))}}{1+e^{-2Q}}\right) \tag{A1}$$

$$h_d(t) = f(\Delta\theta(t)) + \frac{A}{\beta}\left(-1 + 2\frac{e^{-\beta(Q-\Delta\theta(t))}}{1+e^{-2Q}}\right) \tag{A2}$$

where $Q$ is the maximum twist angle, and $f(\Delta\theta(t))$ is the stiffness curve, which can be obtained by using the following expression:

$$f(\Delta\theta) = b_1\Delta\theta + b_2\Delta\theta^3 + b_3\Delta\theta^5 \tag{A3}$$

where $b_1$, $b_2$, $b_3$, $A$, and $\beta$ are constants to be determined.

## Appendix B. Zhang's Hysteresis Model

The total deformation of the HD can be calculated as:

$$\Delta\theta = \frac{\arctan(C_F T_F)}{C_F K_{F0}} - \frac{sign(T_w)}{C_w N K_{w0}}\left(1 - e^{-C_w|T_w|}\right) \tag{A4}$$

where $K_{F0}$, $K_{w0}$, $C_w$, and $C_F$ are constants to be determined, $N$ denotes the gear ratio, and $T_w$ incorporates information about the direction-dependent friction torque.

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
