# Peer review of "Modeling of Transmission Compliance and Hysteresis Considering Degradation in a Harmonic Drive"

_applsci, doi:10.3390/app11020665_

Round 1

Reviewer 1 Report

Please see attached pdf document for my comments. My main recommendation is to create a single comprehensive GLOSSARY/LIST OF SYMBOLS for the paper. 

Reviewer 2 Report

The article focuses on the proposal of a model that captures transmission compliance and hysteresis as well as degradation associated with a harmonic drive (HD). The authors have investigated the existing models in-depth and proposed a new model that can possibly enhance the performance of the system. Some clarifications are required from authors before the article can be accepted for final production. Below are my queries:

  1. I feel that the figures can be placed right below the text where they talk about it. For example, when the authors discuss about the results of their proposed model, they put in phrases about Figure 11, 12 etc but they are being organized on a separate page. It would be better if they can rearrange immediately after the texts/phrases to ensure good continuity. This could be done for all figures.
  2. In line 117: the authors provide equations of piecewise linear function based on catalog reference [5]. However, the link provided in bibliography points to a big list of HD’s. Could they provide the exact link of the model considered? It would add more information to the readers.
  3. Eqs. 22 and 23 have no connection with the phrases? Could the authors add more information on how the constants are changed from a to a’ and b to b’?
  4. Could the authors add more information on how they arrived to Eqs. 24 and 25?
  5. Eq. 37: This equation is probably the objective function for the optimization problem. The authors have not discussed in detail how they did the optimization approach, the methodology/tool, the constraints (if any)? This information is important as the authors speak about optimization only in the abstract.

Reviewer 3 Report

The harmonic drive is studied by a new model considering the transmission compliance and hysteresis and the degradation of teeth is analysed. The identification of parameters is described together with the test rig.

The core of the paper is the identification described in expression (24).

The paper is interesting, but I have some issue:

-Can you evaluate the difference between results from the  model in percentage terms?I mean  about  simulation results of the local elastic coefficient.

  -Figure 5 is superfluous

-English have to be revised
